# mmu-miRNA-342-3p promotes hepatic stellate cell activation and hepatic fibrosis induced by *Echinococcus multilocularis* infection via targeting Zbtb7a

Shanling Cao[1,2], Dexian Wang[1], Yixuan Wu[1], Junmei Zhang[1,2], Lixia Pu[1], Xuenong Luo[1], Xueyong Zhang[1,3], Xiaolin Sun[2], Yadong Zheng[4]*, Shuai Wang[1,5]*, Xiaola Guo[1]*

**1** State Key Laboratory for Animal Disease Control and Prevention, Key Laboratory of Veterinary Parasitology of Gansu Province, Lanzhou Veterinary Research Institute, Chinese Academy of Agricultural Sciences, Lanzhou, Gansu, China, **2** College of Veterinary Medicine, Gansu Agricultural University, Lanzhou, China, **3** Qinghai Academy of Animal Sciences and Veterinary Medicine, Qinghai University, Xining P. R. China, **4** Key Laboratory of Applied Technology on Green-Eco-Healthy Animal Husbandry of Zhejiang Province, Zhejiang Provincial Engineering Laboratory for Animal Health Inspection & Internet Technology, Zhejiang International Science and Technology Cooperation Base for Veterinary Medicine and Health Management, China-Australia Joint Laboratory for Animal Health Big Data Analytics, College of Animal Science and Technology & College of Veterinary Medicine of Zhejiang A&F University, Hangzhou, China, **5** State Key Laboratory for Animal Disease Control and Prevention, College of Veterinary Medicine, Lanzhou University, Lanzhou, Gansu, China

* zhengyadong@zafu.edu.cn (YZ); wangshuai@caas.cn (SW); guoxiaola@caas.cn (XG)

## Abstract

Liver fibrosis is one of the histopathological characters during *Echinococcus multilocularis* infection. The activation of hepatic stellate cells (HSCs) is a key event in the development of liver fibrosis. However, the molecular mechanism of HSC activation in the *E. multilocularis* infection-induced liver fibrosis remains largely unclear. Here, we reported that mmu-miR-342-3p was most dominantly expressed in HSCs and was upregulated in the HSCs in response to *E. multilocularis* infection. We further showed that mmu-miR-342-3p was able to bind to the 3' UTR of the *Zbtb7a* gene and regulated its expression. Moreover, mmu-miR-342-3p expression was negatively correlated with its target gene *Zbtb7a* in HSCs during *E. multilocularis* infection. Knockdown of mmu-miR-342-3p promoted the expression of *Gfap* in the activated HSCs *in vitro*. In the *E. multilocularis*-infected mice, knockdown of mmu-miR-342-3p suppressed the expression of *α-Sma*, *Col1α1*, and *TGF-β* but promoted the expression of *Gfap*. Therefore, mmu-miR-342-3p is a key regulator for activation of HSCs, and inhibiting mmu-miR-342-3p to suppressed Zbtb7a-mediated TGF-β signaling in activated HSCs could be a novel strategy to treat liver fibrosis induced by *E. multilocularis*.

## Author summary

Liver fibrosis is one of the histopathological features during *Echinococcus multilocularis* infection. The activation of hepatic stellate cells (HSCs) is a key event in the development

**Data Availability Statement:** All relevant data are within the paper and its Supporting Information files.

**Funding:** The study was financially supported by grants from the National Key Research and Development Program of China (2022YFD1800200) and the National Natural Science Foundation of China (32273031) to XG, Scientific Research and Development Talent Fund of Zhejiang Agriculture and Forestry University (2021LFR038) to YZ, and State Key Laboratory for Animal Disease Control and Prevention (SKLVEB2020KFKT004) to XZ. The funders had no role in study design, data collection and analysis, decision to publish, or preparation of the manuscript.

**Competing interests:** The authors have declared that no competing interests exist.

of liver fibrosis. However, the molecular mechanism of HSC activation in the liver fibrosis induced by *E. multilocularis* infection remains largely unknown. In recent years, there is increasing evidence indicate that the abnormal expression of miRNAs is closely related to the occurrence and development of liver fibrosis. The expression profile of miRNA in hepatic fibrosis induced by *E. multilocularis* infection has been widely characterized, but the roles of miRNAs in liver fibrosis are largely unexplored. Herein, we identified a set of differentially expressed miRNAs in the activated HSCs induced by *E. multilocularis*, mmu-miR-342-3p of which was dominantly expressed in HSCs. This study signifies the important role of mmu-miR-342-3p in the HSC activation, which will help us to better understand the mechanism of liver fibrosis induced by *E. multilocularis* infection.

## Introduction

Alveolar echinococcosis (AE), a serious zoonotic parasitic disease, is characterized by continuous and infiltrative tumor-like invasive growth of *Echinococcus multilocularis* metacestodes [1]. Humans are occasionally infected by ingesting eggs from contaminated food or water. AE is mainly distributed in semi-agricultural and semi-pastoral areas in the northwest of China, which seriously threatens local people's health [2–4].

Hepatic fibrosis is a dynamic pathological process characterized by excessive deposition of extracellular matrix (ECM) during the progression of parasitic liver disease [5]. The eggs released from *Schistosoma mansoni* and *Schistosoma japonicum* primarily deposits in the periportal zones, which generate a granulomatous reaction and causing substantial pathologic liver fibrosis [6]. *Echinococcus* parasites induce an imbalance of the immune responses within the hepatic tissue, leading to the hepatic architectural distortion and the development of fibrosis [7]. The resultant liver fibrosis may be associated with severe pathology in echinococcosis with granuloma formation, collagen accumulation and inflammation [8,9]. The severity degree of liver injury and fibrosis gradually aggravates with the extension of infection period [7,10]. It is well known that activation of hepatic stellate cell (HSC) is linked to the occurrence and development of liver fibrosis and has been well investigated in a number of liver diseases [11–13]. Upon liver damage, the quiescent HSCs are activated and trans-differentiated into myofibroblast-like cells, which secrete a great amount of ECM. *E. multilocularis* infection can also induce the activation of HSCs, which express high levels of alpha-smooth muscle actin (α-SMA), Vimentin, and Collagen I (Col1α1) [14,15], however the mechanism behind remains unclear.

MicroRNAs (miRNAs) are a class of endogenous non-coding small RNAs with 20–24 nt in length [16]. MiRNAs usually bind to the 3' untranslated region (UTR) of target genes and inhibit their translation or promote degradation. There is increasing evidence that miRNAs play an important role in many infectious liver diseases, such as viral, bacterial and parasitic hepatitis [17,18]. The abnormal expression of miRNAs is closely related to the occurrence and development of liver fibrosis. A large number of miRNAs have been proposed as predictors of fibrosis progression [19,20]. The miRNA expression profile of hepatic fibrosis induced by *E. multilocularis* infection have been widely characterized [21,22], but the roles of miRNAs in liver fibrosis are largely unexplored. Herein, we identified a set of differentially expressed miRNAs in the activated HSCs induced by *E. multilocularis*, mmu-miR-342-3p of which was dominantly expressed in HSCs. We investigated the role of mmu-miR-342-3p in the HSC activation, which will help us to better understand the mechanism of liver fibrosis induced by *E. multilocularis* infection.

## Materials and methods

### Ethics statement

Animal experiments in the study were evaluated and approved by Ethics Committee of Lanzhou Veterinary Research Institute, Chinese Academy of Agricultural Sciences and performed in accordance with Good Animal Practice of Animal Ethics Procedures.

### Parasite infection

Four-week-old BALB/c mice (n = 160) were randomly divided into two groups. One group (n = 80) was intraperitoneally injected with 600 *E. multilocularis* protoscoleces, which were obtained from the hydatid cysts in infected mouse in our lab as previously described [23]. The other (n = 80) was inoculated with 0.9% saline solution as an uninfected group. The high-fat diet induced mice were ordered from Wuhan Servicebio Technology Co., Ltd.

### Histological assessment and immunohistochemistry staining

Liver tissue samples of uninfected (n = 3) and infected mice (n = 3) were fixed in 4% paraformaldehyde, dehydrated, and embedded in paraffin. The samples were sliced into 5 μm sections and performed with hematoxylin and eosin, Masson's trichrome and Sirius Red staining. For Sirius Red staining, sections were incubated in Picro-Sirius Red solution (Abcam) for 60 min, followed by two quick washes in acetic acid solution and absolute alcohol. The images were taken using a light microscope (Zeiss, Germany). Images taken from 15 random 20× fields from three different liver lobes in each animal were measured. The amount of collagen deposition was measured by the method of color thresholding segmentation using ImageJ.

Immunohistochemistry staining was performed according to the following protocol. In brief, the sections were deparaffinized in xylene and rehydrated in graded alcohols. Following antigen retrieval in 0.01 M sodium citrate buffer, the tissue slides were incubated with 3% hydrogen peroxide for 5 minutes to block endogenous peroxidase activity. The sections were blocked with 5% BSA in TBST and then incubated with α-SMA antibodies (1:200, Servicebio). After wash, the sections were incubated with the anti-rabbit secondary antibodies (1:1000, SeraCare) for 1 h at 37°C. Finally, the signal was developed utilizing DAB substrate Kit (Servicebio), and the slices were stained with hematoxylin (Servicebio). The images were taken under a light microscope (Zeiss, Germany). Images taken from 15 random 20× fields from three different liver lobes in each animal were measured. The amount of α-SMA positive cells was measured using ImageJ.

### Isolation, cultivation, and transfection of HSCs

Primary HSCs were isolated from uninfected (n = 6) and infected (n = 6) mouse liver by collagenase IV perfusion, followed by density gradient centrifugation as previously described [24]. Briefly, the livers were perfused with 0.09% EGTA buffer at a flow rate of 5 mL/min at 42°C for 2 min and then with Enzyme buffer containing 0.04% collagenase IV at a flow rate of 5 mL/min at 42°C for 6 min. The resultant digested livers were excised, and *in vitro* digestion was performed in 80 mL Enzyme buffer containing 0.08% collagenase IV and 1% DNase at 37°C for 30 min. The cells were passed through nylon filters (70 μm) and centrifuged at 50 g at 4°C for 4 min. The supernatant was centrifuged at 600 g for 10 min, and the pellet was washed by Gey's balanced salt solution (GBSS) at 500 g at 4°C for 5min to obtain nonparenchymal cells (NPCs). The NPCs were resuspended in 5 mL GBSS and gently coated with Optiprep solutions of different concentrations, with 8 mL 11.5% Optiprep in the upper layer and 4 mL 20% Optiprep in the lower layer, and centrifuged uninterrupted at 1,400 ×g at 4°C for 17 min.

The upper cells (HSCs) were transferred to tubes and washed three times with GBSS. Cells were cultured in DMEM medium containing 10% FBS for subsequent experiments.

For further validation, we first performed flow cytometry analysis using a HSC marker (*Glail fibrillary acidic protein*, *GFAP*). Next, Vitamin A lipid droplet autofluorescence in isolated primary HSCs was detected at a wavelength of 328 nm using a fluorescence microscope (Zeiss, Germany). Thirdly, the expression of the biomarkers for HSCs (*α-SMA*, *GFAP* and *Col1α1*), hepatocytes (*albumin*, *Alb*; *Cytokeratin 18*, *CK18*), and macrophages (*EGF-like module-containing mucin-like hormone receptor-like 1*, *Emr1*) were determined by qRT-PCR.

Primary HSCs were cultured in DMEM medium containing 10% FBS, and transfection experiments were performed (at 5 days after isolation). HSCs ($5 \times 10^5$ cells per well) were plated into 12-well plates and transfected with 50 nM mmu-miR-342-3p inhibitor or 50 nM negative control (NC) (Ambion/Invitrogen) using Lipofectamine RNAiMAX Transfection Reagent (Invitrogen). Following transfection for 10 h, the medium was replaced with fresh DMEM medium supplemented with 10% FBS. Each transfection was independently repeated three times.

## Identification of differentially expressed known miRNAs

The miRNA expression in HSCs after *E. multilocularis* infection was investigated using high-throughput sequencing [25]. Raw reads were generated subsequently, which were deposited in NCBI Sequence Read Archive (SRA) under accession number PRJNA732233. The process of raw sequencing data was performed according to the previously reported methods with some modification. In brief, after removal of the reads with low quality and adaptor sequences, the clean reads were mapped to the full annotated genome of the mouse genome (http://www.ncbi.nlm.nih.gov/genome/genomes/52). Afterward, all the mapped reads were used for identification of miRNAs (miRBase, http://www.mirbase.org/index.shtml). Relative expression levels of miRNAs in three groups were analyzed using DESeq R package (1.8.3) and differentially expressed miRNAs between the uninfected and infected groups at different time points post infection were identified.

Raw reads were generated subsequently, which were deposited in NCBI Sequence Read Archive (SRA) under accession number PRJNA732233. Differentially expressed miRNAs between the uninfected and infected groups at different time points post infection were identified.

## Quantitative RT-PCR Analysis

For examining the miRNA expression, the first-strand cDNAs were synthesized using an All-in-One miRNA First-Strand cDNA Synthesis Kit (GeneCopoia) and U6 snRNA was selected as an endogenous control. For examining the mRNA expression, the first-strand cDNAs were synthesized using a RevertAid First Strand cDNA Synthesis Kit (Invitrogen) and GAPDH was selected as an endogenous reference gene. The cDNA mixture was diluted by 5-fold with nuclease-free water. Quantitative RT-PCR was performed using All-in-One qPCR Mix (GeneCopoeia) by 7500 Real Time PCR System (Applied Biosystems) under following conditions: 95˚C for 10 min, followed by 40 cycles of 95˚C for 10 sec, 60˚C for 1min. All the primers were purchased from GeneCopoeia (Table A in S1 Text). The relative expression levels of miRNAs or mRNA were calculated using the $2^{-\Delta\Delta Ct}$ formula. Statistical analysis data were taken from three independent experiments.

## Plasmid construction and luciferase assay

The 3' UTR fragment of Zinc finger and BTB domain-containing 7A (*Zbtb7a*) gene with restriction enzyme site was amplified using a pair of primers: 5'-

GAGCTCGGTGAATTTGCGTGT-3' and 5'-GTCGACCCTGTGTCCCTCCTA-3' (restriction enzyme sites were underlined). The PCR product was cloned into a PmirGLO Dual-Luciferase vector (Promega, USA) and confirmed by sequencing, designated as WT-*Zbtb7a*. The 3'UTR of *Zbtb7a* with mutations in a binding site was artificially synthesized (Sangon, China), cloned into the PmirGLO Dual-Luciferase vector and confirmed by sequencing, designated as Mut-*Zbtb7a*.

HEK293T cells ($1 \times 10^5$ cells per well) were plated into 24-well plates and transfected with 1μg of WT-*Zbtb7a* or Mut-*Zbtb7a* in combination with 30 pmol of mmu-miR-342-3p mimics or NC mimics using Lipofectamine 2000 (ThermoFisher Scientific). At 24 h after transfection, Renilla and firefly luciferase activity was measured by GloMax 96 Microplate Luminometer (Promega) using Dual-Glo Luciferase Assay System (Promega). Each transfection was independently repeated three times.

## Western blotting

The total proteins were isolated from cells using RIPA buffer (ThermoFisher Scientific) supplemented with protease inhibitor (Sigma). The concentration of total proteins was measured using BCA protein assay Kit (ThermoFisher Scientific). A total of 50 μg proteins were separated by SDS-PAGE and transferred onto PVDF membranes (Millipore). Subsequently, the membranes were blocked with 5% BSA in TBST and then incubated with Zbtb7a antibodies (1:500, BBI) at 4˚C overnight. After wash, the membranes were incubated with anti-rabbit secondary antibodies (1:1000, Abcam) for 1 h at room temperature, followed by addition of Pierce ECL Western Blotting Substrate (ThermoFisher Scientific). Light emission was recorded using a chemiluminescent detection system (G-Box, Syngene).

## In vivo study

Chitosan (CS)-miRNA nanoparticles were prepared according to the method previously described with minor modifications [26]. Briefly, 95% deacetylated CS was dissolved in acetic acid solution (2.0 mg/mL, adjust to pH 5.5). A sodium tripolyphosphate solution (TPP; 2.0 mg/mL) was used as a cross-linking agent. To prepare CS-miRNA nanoparticles, 250 μL of the TPP solutions containing 3 nmol miRNA inhibitors were added drop by drop to 2.5 mL of CS solutions and the resulting solutions were maintained at room temperature for 30 min.

Eight *E. multilocularis*-infected mice were randomly divided into two groups and injected with 100 μL of CS-NC nanoparticles (CSNP-NC) or CS-mmu-miR-342-3p inhibitor nanoparticles (CSNP- inhibitor) through the tail vein, respectively. After 40 h, mice were humanely killed and liver was collected for further use.

## Statistical analysis

All statistical analyses were performed using GraphPad Prism 8 (La Jolla, USA), and differences were compared using a two-tailed unpaired *t*-test for two groups and ANOVA for three groups. A $p < 0.05$ was considered as statistically significant.

# Results

## Activation of HSCs and hepatic fibrosis during *E. multilocularis* infection

Histopathological analysis using hematoxylin and eosin, Masson's trichrome, and Sirius Red staining revealed the excessive deposition of collagen surrounding the metacestodes in the liver of *E. multilocularis*-infected mice, being higher than that of control mice and high-fat diet induced mice (Figs 1A and S1). The results of α-SMA immunostaining showed that

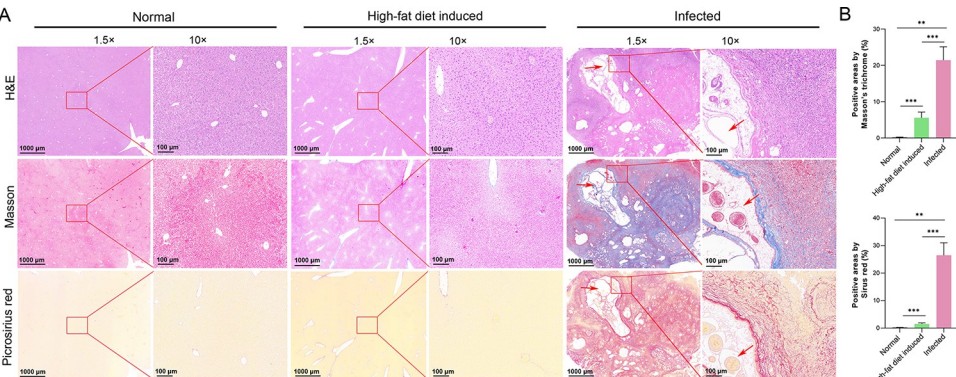

**Fig 1. The effects of *E. multilocularis* infection on hepatic fibrosis in mouse.** (A) Representative pictures of Hematoxylin and Eosin (H&E), Masson's Trichrome, and Picrosirius Red staining for liver fibrogenesis in control mice, high-fat diet induced mice, and *E. multilocularis*-infected mice. Arrows indicate cysts. (B) The birefringent collagen content was quantified in Masson's Trichrome- or Picrosirius Red-stained sections by color thresholding-based segmentation using ImageJ.

myofibroblastic HSCs (MF-HSCs) mainly distributed in the livers of *E. multilocularis*-infected mice around the metacestodes (Fig 2A). The proportion of MF-HSCs (α-SMA-positive cells) and macrophages (Emr1-positive cells) in the livers of *E. multilocularis*-infected mice were higher than those in un-infected group, whereas hepatocytes (CK18-positive cells) were significantly decreased (Figs 2B and S2). To assess the effect of *E. multilocularis* infection on the HSC activation, the primary HSCs were isolated and the expression of biomarkers for HSCs were examined. Flow cytometry analysis demonstrated that the isolated HSCs accounted for approximately 90% (Fig 2D) and exhibited a striking blue autofluorescence (Fig 2C). Furthermore, the isolated HSCs that highly expressed three biomarkers for HSCs (determined by *α-SMA*, *GFAP* and *Col1α1*) reduced the contamination with hepatocytes (determined by *Alb* and *CK18*) and macrophages (determined by *Emr1*) (S3 Fig). As expected, the expression of *α-Sma*, *Col1α1*, and *Vimentin* were significantly upregulated 30-, 60-, 90-day post infection compared with the uninfected group (Fig 2E). Increased expression of α-Sma at protein level was observed in HSCs from *E. multilocularis*- infected mice 30-, 60-, and 90-day post infection (Fig 2F). These results demonstrate that the hepatic fibrosis and HSC activation occur in response to *E. multilocularis* infection.

## Dysregulation of mmu-miR-342-3p in HSCs during *E. multilocularis* infection

Considering the important roles of miRNAs in HSC activation, we recently characterized the global profiling of lncRNAs-miRNAs-mRNAs in hepatocytes (HCs), Kupffer cells (KCs) and HSCs of mouse liver during *E. multilocularis* infection [27]. We found that 49 and 67 miRNAs were differentially expressed in the HSCs 60- and 90-day post infection, respectively (Table B and C in S1 Text). Of them, 33 were commonly shared (Fig 3A). The KEGG pathway analysis indicated that target genes of these miRNAs were mainly involved in cancer pathway, Hippo signaling pathway, TGF-β signaling pathway, and PI3K-Akt signaling pathway (Fig 3B). Among these differentially expressed miRNAs, mmu-miR-342-3p was dominantly expressed in HSCs compared with that in HCs and KCs ($p < 0.001$, Fig 3C). Thus, we analyzed the dynamic expression of mmu-miR-342-3p in HSCs at different time points post infection. The qRT–PCR analysis showed that the expression of mmu-miR-342-3p was significantly

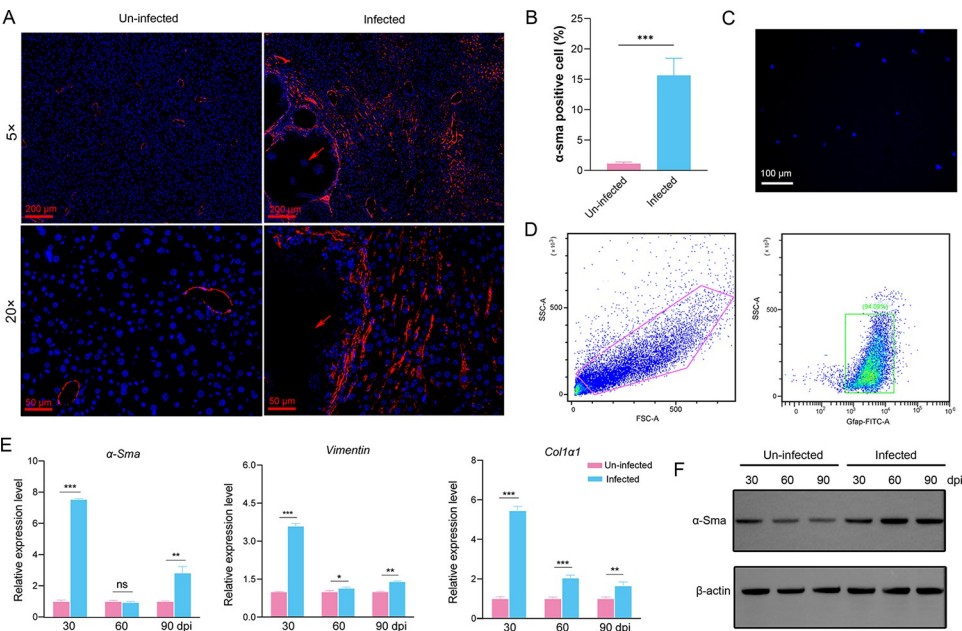

**Fig 2. The activation of HSCs induced by *E. multilocularis* infection.** (A) Representative pictures of immunofluorescence staining for α-SMA in the livers of *E. multilocularis*-infected and uninfected mice. Arrows indicate cysts. (B) Quantification of α-SMA-positive cells in the livers of *E. multilocularis*-infected and uninfected mice. (C) Vitamin A lipid droplet autofluorescence in isolated primary HSCs was detected at a wavelength of 328 nm using a fluorescence microscope. (D) Flow cytometry analysis of isolated mouse HSCs using Gfap. (E) The expression of *α-sma*, *col1α1*, and *vimentin* in the HSCs from *E. multilocularis*-infected mice 30-, 60-, and 90-day post-infection (dpi) by qRT-PCR. (F) The expression of α-SMA in the HSCs from *E. multilocularis*-infected mice 30-, 60-, and 90-day post-infection (dpi) by Western blotting. Data for final statistical analysis were taken from 3 independent experiments. $^*p < 0.05$, $^{**}p < 0.01$, $^{***}p < 0.001$.

upregulated in the HSCs from *E. multilocularis*-infected mice 30- and 60-day post infection, while its expression was decreased 90-day post infection ($p < 0.001$, Fig 3D).

## Upregulation of mmu-miR-342-3p in the culture-activated HSCs

As primary HSCs are known to be activated during cultivation [28], the quiescence HSCs were isolated and cultured in plastic plates for 9 days. As expected, the expression of *α-Sma* and *Col1α1* was significantly up-regulated at day 9 compared with that at day 0, while the expression of *Gfap*, a biomarker for HSC quiescence, was significantly downregulated ($p < 0.01$, Fig 4A). Consistently, the immunofluorescence staining showed that α-SMA protein was highly expressed in culture-activated HSCs (at day 9), suggesting that the primary HSCs are activated (Fig 4B). Next, we assessed whether the expression of mmu-miR-342-3p was changed in the culture-activated HSCs. We found that it was significantly upregulated at day 9) compared with that at day 0 (Fig 4C). The results suggest that the mmu-miR-342-3p may be involved in the HSC activation.

## Inhibition of Zbtb7a by mmu-miR-342-3p via directly binding to its 3′UTR

To determine the potential role of mmu-miR-342-3p in HSC activation during *E. multilocularis* infection, its potential targets were identified by using TargetScan, PicTar and miRNA. org. A total of 14 commonly shared targets were screened out, which were mainly involved in cell cycle, cell differentiation, cell proliferation, and tumorigenesis (Table D in S1 Text). Among these candidates, Zbtb7a was first selected for downstream analysis, of which contains

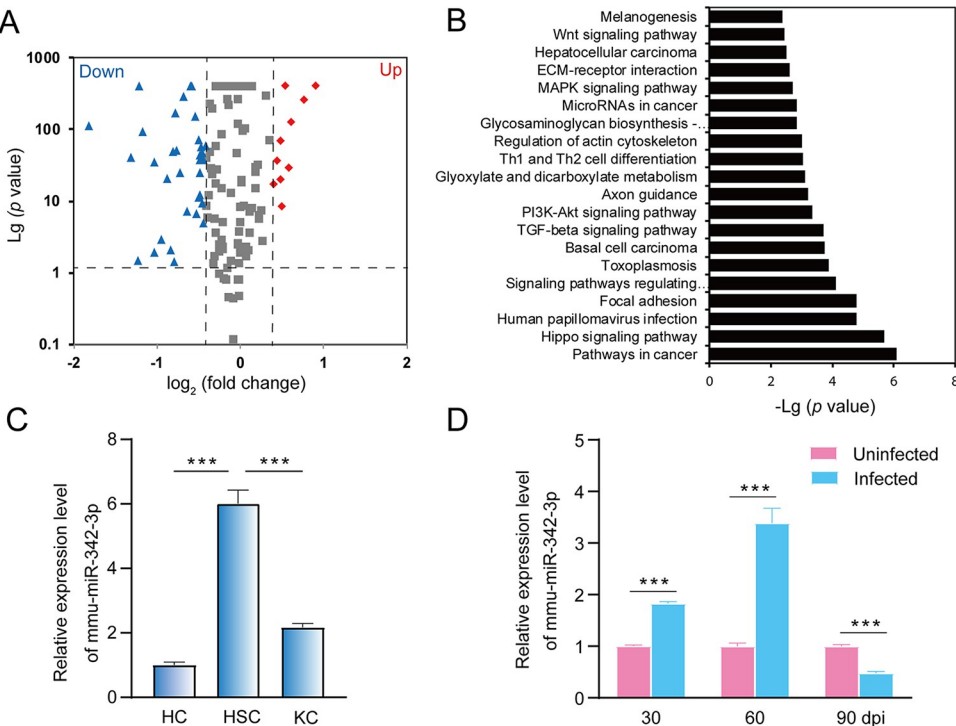

**Fig 3. Dysregulation of mmu-miR-342-3p in liver HSCs during *E. multilocularis* infection.** (A) A volcano plot of the differentially expressed miRNAs between two liver HSCs from *E. multilocularis*-infected mice at 60 and 90-day post infection (dpi), where the red and green, indicated significantly upregulated miRNAs (Up) and downregulated miRNAs (Down), respectively. (B) KEGG pathway enrichment analysis of target genes. (C) The expression of mmu-miR-342-3p in hepatocyte cells (HCs), Kupffer cells (KCs), and hepatic stellate cells (HSCs). (D) The expression of mmu-miR-342-3p in liver HSCs from *E. multilocularis*-infected mice at 30-, 60-, and 90-day post-infection (dpi). The uninfected mice at each sampling time point were used as controls. Data for final statistical analysis were taken from 3 independent experiments. ***$p < 0.001$.

a putative binding site for mmu-miR-342-3p (Fig 5A). To validate the binding capability, the WT-Zbtb7a-3′UTR and the Mut-Zbtb7a-3′-UTR luciferase reporter systems were constructed and co-transfected with mmu-miR-342-3p mimics or NC into HEK293T cells, respectively. Luciferase reporter assay revealed that the mmu-miR-342-3p mimic significantly decreased the luciferase activity in WT-*Zbtb7a*-3′UTR-transfected HEK293T cells compared with that in the control ($p < 0.001$, Fig 5A). However, the decrease was not observed in the Mut-*Zbtb7a*-3′-UTR-transfected cells ($p > 0.05$, Fig 5A), suggesting that mmu-miR-342-3p is able to bind to the *Zbtb7a*-3′-UTR. Consistently, after downregulation of mmu-miR-342-3p in the HSCs from *E. multilocularis*-infected mice by transfecting with mmu-miR-342-3p inhibitor ($p < 0.05$, Fig 5B), Zbtb7a was significantly up-regulated at both mRNA and protein levels (Fig 5C). Moreover, the expression of *Zbtb7a* was negatively correlated with the mmu-miR-342-3p expression in the HSCs from *E. multilocularis*-infected mice ($p = 0.0031$, Fig 5D and E). These results suggest that mmu-miR-342-3p participates in the HSC activation via targeting Zbtb7a.

## The *in vivo* effect of down-regulated mmu-miR-342-3p on HSC activation during *E. multilocularis* infection

In order to study the effect of mmu-miR-342-3p expression on the HSC activation *during E. multilocularis* infection, the primary HSCs were isolated from *E. multilocularis*-infected mice.

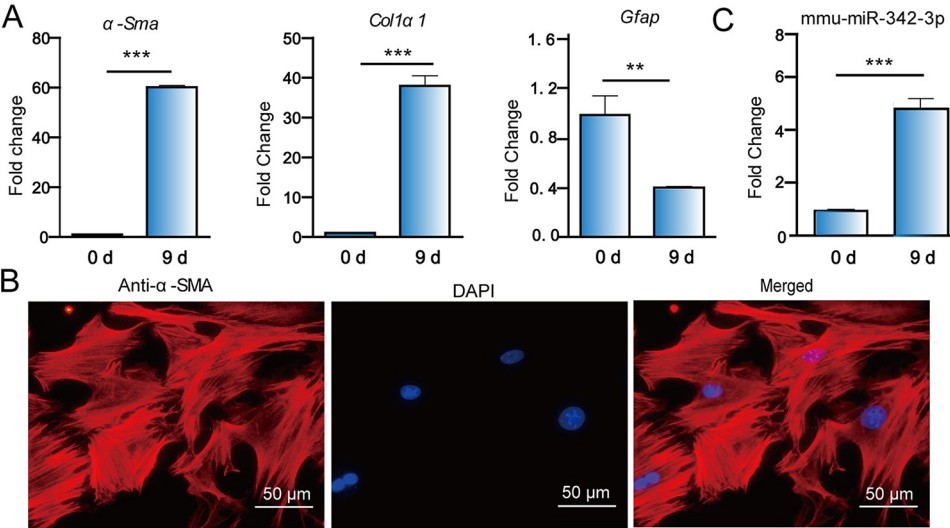

**Fig 4. Upregulation of mmu-miR-342-3p in the culture-activated HSCs.** (A) The qRT-PCR analysis of *α-Sma*, *Col1α1*, *Gfap* in *in vitro* culture-activated HSCs. (B) Immunofluorescence staining of α-SMA in *in vitro* culture-activated HSCs. Scale bar = 50 μm (C) The expression of mmu-miR-342-3p in *in vitro* culture-activated HSCs. Data for final statistical analysis were taken from 3 independent experiments. **$p < 0.01$, ***$p < 0.001$.

We found that downregulated mmu-miR-342-3p could significantly promote the expression of *Gfap*, while *α-Sma* and *Vimentin* were not significantly changed (Fig 6A and 6B). In order to *in vivo* assess the effect of mmu-miR-342-3p on hepatic fibrosis, CS-mmu-miR-342-3p

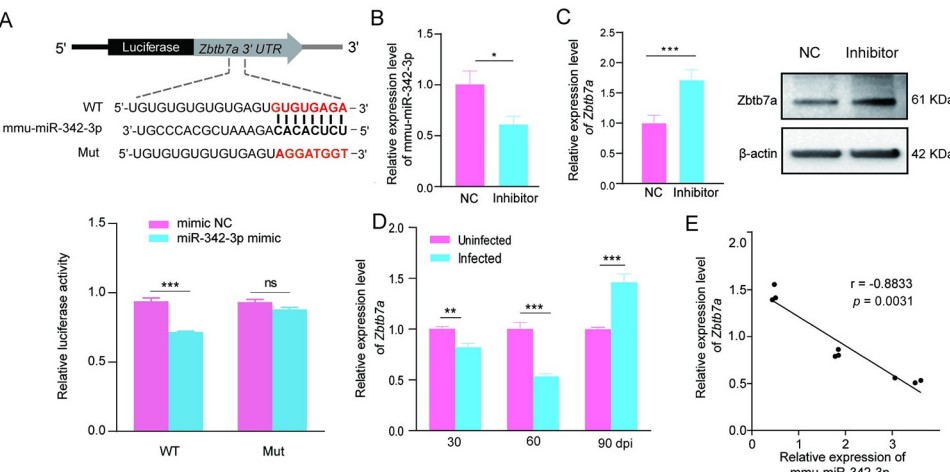

**Fig 5. Inhibition Zbtb7a by mmu-miR-342-3p via directly binding to its 3′UTR.** (A) Schematic representation of a putative binding site of mmu-miR-342-3p in the 3′UTR of *Zbtb7a* (up) and the dual-Glo Luciferase assay for the interaction between mmu-miR-342-3p-Mimics and *Zbtb7a* (down). The miRNA 'seed' sequences are indicated in bold in black. WT: WT- *Zbtb7a* or Mut- *Zbtb7a* construct was co-transfected into 293T cells with either mimic negative control (mimic NC) or mmu-miR-342-3p mimic (miR-342-3p mimic) and luciferase activities were measured after 24 h transfection. (B) The expression of mmu-miR-342-3p in the HSCs from *E. multilocularis*-infected mice transfected with inhibitor negative control (NC) or mmu-miR-342-3p inhibitor (Inhibitor) by RT-qPCR. (C) The expression of Zbtb7a at mRNA and protein levels in the HSCs from *E. multilocularis*-infected mice transfected with mmu-miR-342-3p inhibitor (Inhibitor). (D) The expression of *zbtb7a* in primary HSCs from *E. multilocularis*-infected mice at 30-, 60-, and 90-day post-infection (dpi). (E) Correlation between the expression of *Zbtb7a* and mmu-miR-342-3p. (n = 9, Spearman's correlation analysis; r, correlation coefficient) Data for final statistical analysis were taken from 3 independent experiments. **$p < 0.01$; ***$p < 0.001$.

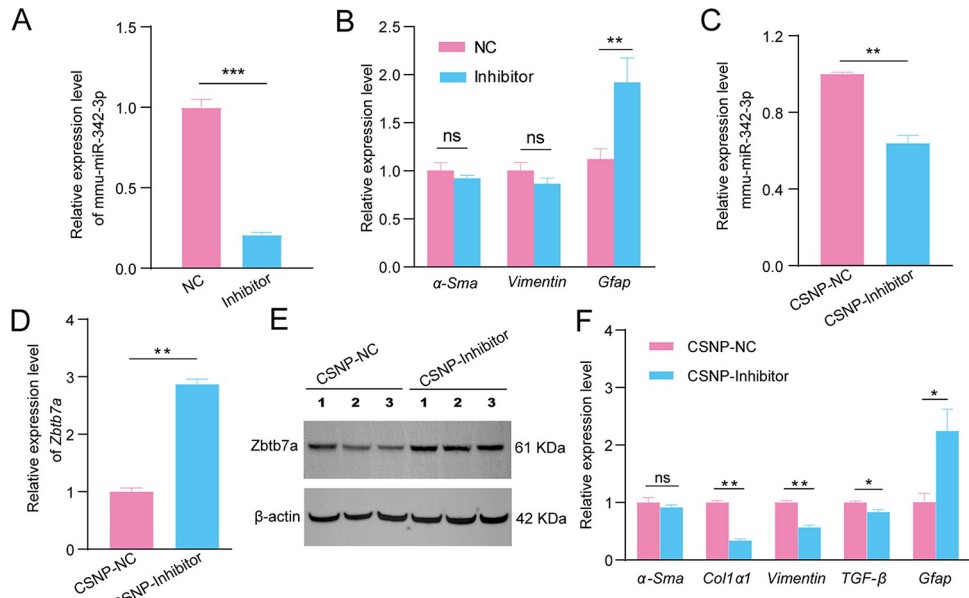

**Fig 6. The effect of down-regulated mmu-miR-342-3p on HSC activation during *E. multilocularis* infection.** (A) The expression of mmu-miR-342-3p in the HSCs from *E. multilocularis*-infected mice after transfecting with inhibitor negative control (NC) or mmu-miR-342-3p inhibitor (inhibitor). (B) Effects of downregulated mmu-miR-342-3p on the expression of HSC activation marker (*α-Sma* and *Vimentin*) and quiescent marker *Gfap* in the HSCs from *E. multilocularis*-infected mice. (C) The mmu-miR-342-3p expression was significantly downregulated in the mouse livers by CS-mmu-miR-342-3p inhibitor nanoparticles (CSNP-Inhibitor) compared with that of CS-inhibitor negative control nanoparticles (CSNP-NC). (D) The Zbtb7a expression both at mRNA and protein levels were significant upregulated in the livers of *E. multilocularis*-infected mice by injecting with CS-mmu-miR-342-3p inhibitor nanoparticles. (F) Effect of mmu-miR-342-3p on the expression of *α-Sma*, *Col1α1*, *Vimentin*, and TGF-β in the in the livers of *E. multilocularis*-infected mice by injecting with CS-mmu-miR-342-3p inhibitor nanoparticles. Data for final statistical analysis were taken from 3 independent experiments. ***$p < 0.001$.

nanoparticles were injected into mice through the tail vain. As expected, mmu-miR-342-3p was found to be downregulated ($p < 0.01$, Fig 6C) and its target gene Zbtb7a was remarkably increased at both mRNA and protein levels in the liver of mice injected with CSNP-mmu-miR-342-3p inhibitor ($p < 0.01$, Fig 6D and 6E). Moreover, *Col1α1*, *Vimentin* and *TGF-β* were significantly downregulated, while *Gfap* was significantly upregulated ($p < 0.05$, Fig 6F). These results suggest that the parasite infection induces HSC activation via the mmu-miR-342-3p-Zbtb7a axis.

## Discussion

Liver fibrosis is a significant histological hallmark of AE and the activation of HSCs is a central link to the initiation and progression of hepatic fibrosis [15]. HSCs are mainly activated and rapidly transform into proliferative, migratory, and extracellular matrix-producing myofibro-blasts [29]. In this study, we found hepatic fibrosis occurred in response to *E. multilocularis* infection, characterized by the excessive deposition of collagen in the liver surrounding the metacestodes. In line with this, we also found that the higher proportion of myofibroblastic HSCs as well as higher expression of *Vimentin*, *Col1α1*, and *α-Sma* in liver HSCs from AE mice. Our results confirmed that the HSCs were activated in response to *E. multilocularis* infection, thus promoted the process of liver fibrosis.

Liver fibrogenesis is regulated by multiple growth factors and cytokines. Among those regu-lators, the best known prosclerotic mediator is TGF-β family [11]. MiRNAs, known as small

regulatory RNAs, have recently emerged as critical regulatory molecules in chronic liver diseases [30]. Many studies have shown that the abnormal expression of miRNAs is closely associated with the HSC activation during the development of liver fibrosis [18]. In this study, we identified a large number of aberrantly expressed miRNAs in the HSCs in response to *E. multilocularis* infection. As expected, some of them including mmu-miR-29, mmu-miR-192 and mmu-miR-378 have been reported to be associated with the activation of HSCs [31]. To understand the regulatory role of these differentially expressed miRNAs, the target genes were predicted. Among them, some were involved in the TGF-β signaling pathway and PI3K-Akt signaling pathway. For instance, miR-29 is highly expressed in HSCs and possibly inhibits the HSC activation by targeting the PI3K/AKT signaling pathway [31]. Similarly, *S. japonicum* miR-29b-3p prevents parasite-induced liver fibrosis by inhibiting COL1A1 and COL3A1[32]. miR-192 is highly expressed in quiescent HSCs and can suppress HSC activation by targeting TGF-β1/Smad signaling [33].

In the study, mmu-miR-342-3p was proved to be dominantly expressed in HSCs. We also observed its increased expression in the HSCs from *E. multilocularis*-infected mouse liver at 30- and 60-day post infection. These results suggest mmu-miR-342-3p to be a promising candidate miRNA affecting the progression of liver fibrosis induced by *E. multilocularis* infection. Next, we found that mmu-miR-342-3p was significantly upregulated *in vitro* culture-activated HSCs, suggesting its potential role in the HSC activation. Luciferase reporter assay revealed that the mmu-miR-342-3p mimic could directly bind to *Zbtb7a*. Moreover, we observed upregulation of Zbtb7a in activated HSCs transfected with mmu-miR-342-3p inhibitor. Consistently, the expression of *Zbtb7a* was negatively correlated with the mmu-miR-342-3p expression in the HSCs from *E. multilocularis*-infected mice. These results indicate that mmu-miR-342-3p can repress Zbtb7a expression by directly targeting its 3'-UTR. It has been shown that hypermethylation of Zbtb7a is associated with the liver fibroblast activation and fibrosis [34]. Zbtb7a can inhibit the expression of TGF-β1 through indirectly suppressing its promoter activity of TGF-β1 [34]. As the most potent fibrogenic factor, TGF-β is the primary factor in the transformation of HSCs to myofibroblasts [33,34]. TGF-β exerts its actions through binding to serine/threonine kinase transmembrane TGF-β receptor I and II, resulting in deposition of extracellular matrix (ECM) proteins, such as Col1α1, α-SMA, and fibronectin [35]. Col1α1 is the main component of the fibrotic liver and its expression is known to be induced by TGF-β [36]. We found that down-regulated mmu-miR-342-3p decreased the expression of fibrogenic genes including *Col1α1* and *Vimentin* and up-regulated the quiescence-associated gene *Gfap*. It is possible that mmu-miR-342-3p may promote the HSC activation by targeting Zbtb7a-mediated TGF-β signaling during *E. multilocularis* infection.

## Supporting information

**S1 Data. Excel spreadsheet containing, in separate sheets, the underlying numerical data and statistical analysis for Figure panels 1B, 2B, 2E, 3C, 3D, 4A, 4C, 5A, 5B, 5C, 5D, 6A, 6B, 6C, 6D, 6F and S3.**
(XLSX)

**S1 Fig. Full-blown liver lesions of *E. multilocularis*-infected mice were observed.**
(TIF)

**S2 Fig. Immunostaining of CLEC4F (KC marker) and Cytokeratin 18 (HC marker) in livers of un-infected mice or *E. multilocularis*-infected mice.**
(TIF)

**S3 Fig. The mRNA expression levels of biomarkers in the isolated HCs (*Alb* and *CK18*), HSCs (*α-SMA*, *Gfap*, and *Col1α1*), and KCs (*Emr1*) by RT-qPCR.**
(TIFF)

**S1 Text.** Table A. The qRT-PCR primers used in the study. Table B. Summary of the differentially expressed miRNAs in liver HSCs in *E. multilocularis*-infected mice at 60-day post infection. Table C. Summary of the differentially expressed miRNAs in liver HSCs in *E. multilocularis*-infected mice at 90-day post infection. Table D. Putative target genes of mmu-miR-342-3p.
(DOCX)

## Author Contributions

**Conceptualization:** Yadong Zheng, Xiaola Guo.

**Formal analysis:** Shanling Cao, Xiaola Guo.

**Funding acquisition:** Yadong Zheng, Xiaola Guo.

**Investigation:** Shanling Cao, Dexian Wang, Yixuan Wu, Junmei Zhang.

**Methodology:** Shanling Cao.

**Resources:** Xuenong Luo, Xueyong Zhang.

**Software:** Shanling Cao, Lixia Pu.

**Supervision:** Yadong Zheng, Xiaola Guo.

**Validation:** Shanling Cao.

**Visualization:** Xiaola Guo.

**Writing – original draft:** Shanling Cao, Yadong Zheng, Xiaola Guo.

**Writing – review & editing:** Xiaolin Sun, Yadong Zheng, Shuai Wang, Xiaola Guo.

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
