## [Decision Letter · Decision Letter 0]

21 Feb 2023

Dear Dr Xiaola Guo,

Thank you very much for submitting your manuscript "mmu-miRNA-342-3p promotes hepatic stellate cell activation and hepatic fibrosis induced by Echinococcus multilocularis infection via targeting Zbtb7a" for consideration at PLOS Neglected Tropical Diseases. As with all papers reviewed by the journal, your manuscript was reviewed by members of the editorial board and by several independent reviewers. In light of the reviews (below this email), we would like to invite the resubmission of a significantly-revised version that takes into account the reviewers' comments. 

We cannot make any decision about publication until we have seen the revised manuscript and your response to the reviewers' comments. Your revised manuscript is also likely to be sent to reviewers for further evaluation.

Sincerely,

Alessandra Morassutti, PhD

Academic Editor

Eva Clark

Section Editor

Reviewer's Responses to Questions

**Key Review Criteria Required for Acceptance?**

**Methods**

-Are the objectives of the study clearly articulated with a clear testable hypothesis stated?

-Is the study design appropriate to address the stated objectives?

-Is the population clearly described and appropriate for the hypothesis being tested?

-Is the sample size sufficient to ensure adequate power to address the hypothesis being tested?

-Were correct statistical analysis used to support conclusions?

-Are there concerns about ethical or regulatory requirements being met?

Reviewer #1: (No Response)

Reviewer #2: see below

Reviewer #3: the objectives of the study clearly articulated. however, the authors need to answer the following questions:

Major issues:

• A copy of ethical approval is needed.

• In line 78, it’s better to mention the total number of mice used. And how it’s grouped because here you mentioned you divided them into 2 group, then in line 98 you said that “Relative expression levels of miRNAs in three groups” ???!

• In line 83, more details about the method of isolating HSCs from mice liver is needed.

• In line 133, the authors had mentioned that the liver was fixed in 4% paraformaldehyde, is the paraformaldehyde is prepared in Phosphate buffer saline or what, and is this protocol for immunochemistry does not require antigen retrieval?

Minor Issues:

Clarification of abbreviations is needed to provide better understanding especially for those who are not very familiar with the field of the study.

**Results**

-Does the analysis presented match the analysis plan?

-Are the results clearly and completely presented?

-Are the figures (Tables, Images) of sufficient quality for clarity?

Reviewer #1: (No Response)

Reviewer #2: see below

Reviewer #3: the results was clearly presented.

**Conclusions**

-Are the conclusions supported by the data presented?

-Are the limitations of analysis clearly described?

-Do the authors discuss how these data can be helpful to advance our understanding of the topic under study?

-Is public health relevance addressed?

Reviewer #1: (No Response)

Reviewer #2: see below

Reviewer #3: the conclusions is supported by the data. however the author should mention the limitations of the study clearly.

**Editorial and Data Presentation Modifications?**

Reviewer #1: (No Response)

Reviewer #2: see below

Reviewer #3: “Minor Revision”

**Summary and General Comments**

Reviewer #1: The manuscript entitled "mmu-miRNA-342-3p promotes hepatic stellate cell activation and hepatic fibrosis induced by Echinococcus multilocularis infection via targeting Zbtb7a" has been reviewed. Generally, the manuscript clearly indicated the role of miRNA-342-3p in liver fibrosis associated with Echinococcus multilocularis infection.

The reviewer has several suggestions and comments to improve the quality of current manuscript.

1. Liver fibrosis could be caused by other flatworm infection such as Schistosoma. The authors should include related studies in section of introduction and discussion to enrich the informative manuscript for understanding the mechanism of liver fibrosis.

2. The authors globally characterized the profiles of lncRNAs-miRNAs-mRNAs in hepatocytes (HCs), Kupffer cells (KCs) and HSCs of mouse liver during E. multilocularis infection. Did they note that the Zbtb7a has different expression in different time of different type cells? The author could include the related results in the discussion.

3. In Fig 1A, the data indicated that the expression of Alpha-Sma has not different at 60dpi between infection and uninfection. How to explain the results?

4. In Fig 5E, the manuscript is lack of the legend of fig 5E. What mean 1,2,3 in fig 5E?

Reviewer #2: The MS by Cao et al. entitled: “mmu-miRNA-342-3p promotes hepatic stellate cell activation and hepatic fibrosis induced by Echinococcus multilocularis infection via targeting Zbtb7a” is interesting in basics but suffers from major flaws.

The authors state that infection induces fibrosis in liver of mice infected by EM. All of the message of the MS is based on this histological finding. However, this fundamental part of the analysis is badly documented:

1. In the materials and methods section there is no description how the fibrosis was evaluated nor by whom neither by which method.

2. Infection mice with EM is deleterious. In the few pics shown the authors do not show any metacestodes but just some portal field with fibrosis. Fibrosis develops around the metacestode and not in the portal fields. Authors have to show full blown liver lesions, measure the differences of inflammatory infiltrates, and then measure the fibrotic rim (or computer-based and with values that have been used to generate the box-plots: Combining Computed Tomography and Histology Leads to an Evolutionary Concept of Hepatic Alveolar Echinococcosis - PubMed (nih.gov).

3. Definition of stellate cells is poor. Are HSC increased during infection in situ? What is about the KCs and/or hepatocytes?

4. The authors must demonstrate that the effect is really based on this specific kind of infection (e.g. control groups with fat induced liver disease, or hepatitis are missing).

5. In the paper the number of experiments und number of cell cultures and repetition of experiments are not clearly given. This not acceptable.

Minor: It remains a long time unclear whether these experiments have been performed in mice or humans. Authors should consider to shift the experiments in the human system. This is the major health problem, not mice. Cultures of Human stellate cells are available. 

If all this major issue are taken in account I am willing to re-review the MS.

Reviewer #3: I think this paper miRNA-342-3p promotes HSC activation induced by E. multilocularis infection is very important and novel. 

Major issues:

• A copy of ethical approval is needed.

• In line 78, it’s better to mention the total number of mice used. And how it’s grouped because here you mentioned you divided them into 2 group, then in line 98 you said that “Relative expression levels of miRNAs in three groups” ???!

• In line 83, more details about the method of isolating HSCs from mice liver is needed.

• In line 133, the authors had mentioned that the liver was fixed in 4% paraformaldehyde, is the paraformaldehyde is prepared in Phosphate buffer saline or what, and is this protocol for immunochemistry does not require antigen retrieval?

Minor Issues:

Clarification of abbreviations is needed to provide better understanding especially for those who are not very familiar with the field of the study.

PLOS authors have the option to publish the peer review history of their article (what does this mean?). If published, this will include your full peer review and any attached files.

Reviewer #1: No

Reviewer #2: No

Reviewer #3: No
---

## [Editor Report · Decision Letter 1]

25 Apr 2023

Dear Dr Guo,

Thank you very much for submitting your manuscript "mmu-miRNA-342-3p promotes hepatic stellate cell activation and hepatic fibrosis induced by Echinococcus multilocularis infection via targeting Zbtb7a" for consideration at PLOS Neglected Tropical Diseases. As with all papers reviewed by the journal, your manuscript was reviewed by members of the editorial board and by several independent reviewers. In light of the reviews (below this email), we would like to invite the resubmission of a significantly-revised version that takes into account the reviewers' comments. 

We cannot make any decision about publication until we have seen the revised manuscript and your response to the reviewers' comments. Your revised manuscript is also likely to be sent to reviewers for further evaluation.

Sincerely,

Francesca Tamarozzi

Section Editor

Eva Clark

Section Editor
---

## [Decision Letter · Decision Letter 2]

17 May 2023

Dear Dr Guo,

Thank you very much for submitting your manuscript "mmu-miRNA-342-3p promotes hepatic stellate cell activation and hepatic fibrosis induced by Echinococcus multilocularis infection via targeting Zbtb7a" for consideration at PLOS Neglected Tropical Diseases. As with all papers reviewed by the journal, your manuscript was reviewed by members of the editorial board and by several independent reviewers. The reviewers appreciated the attention to an important topic. Based on the reviews, we are likely to accept this manuscript for publication, providing that you modify the manuscript according to the review recommendations. 

It is of particular importance the last point raised by Reviewer #2 is addressed.

Sincerely,

Francesca Tamarozzi

Section Editor

Eva Clark

Section Editor

It is of particular importance the last point raised by Reviewer #2 is addressed.

Reviewer's Responses to Questions

**Key Review Criteria Required for Acceptance?**

**Methods**

-Are the objectives of the study clearly articulated with a clear testable hypothesis stated?

-Is the study design appropriate to address the stated objectives?

-Is the population clearly described and appropriate for the hypothesis being tested?

-Is the sample size sufficient to ensure adequate power to address the hypothesis being tested?

-Were correct statistical analysis used to support conclusions?

-Are there concerns about ethical or regulatory requirements being met?

Reviewer #1: (No Response)

Reviewer #2: Improved.

**Results**

-Does the analysis presented match the analysis plan?

-Are the results clearly and completely presented?

-Are the figures (Tables, Images) of sufficient quality for clarity?

Reviewer #1: (No Response)

Reviewer #2: Improved. However, as I recommeded in my first review I ask the authors again to insert a histological figure of a full blown liver lesion of a mouse after infection as a proof of principle. Up to now, the authors show histology of mice with non characteristic periportal fibrosis; to insert a histological figure of the charcteristic liver lesion after infection is an item that should be easily done since infection rates are high (as have stated the authors); to solve this point an additional supplemental figure is enough. The authors have not answered to this point raised in my previous review.

**Conclusions**

-Are the conclusions supported by the data presented?

-Are the limitations of analysis clearly described?

-Do the authors discuss how these data can be helpful to advance our understanding of the topic under study?

-Is public health relevance addressed?

Reviewer #1: (No Response)

Reviewer #2: Improved

**Editorial and Data Presentation Modifications?**

Reviewer #1: (No Response)

Reviewer #2: Improved.

**Summary and General Comments**

Reviewer #1: (No Response)

Reviewer #2: Improved.

PLOS authors have the option to publish the peer review history of their article (what does this mean?). If published, this will include your full peer review and any attached files.

Reviewer #1: No

Reviewer #2: No

Figure Files:

Data Requirements:

Reproducibility:

References

---

## [Editor Report · Decision Letter 3]

18 Jun 2023

Dear Dr Guo,

Thank you very much for submitting your manuscript "mmu-miRNA-342-3p promotes hepatic stellate cell activation and hepatic fibrosis induced by Echinococcus multilocularis infection via targeting Zbtb7a" for consideration at PLOS Neglected Tropical Diseases. As with all papers reviewed by the journal, your manuscript was reviewed by members of the editorial board and by several independent reviewers. The reviewers appreciated the attention to an important topic. Based on the reviews, we are likely to accept this manuscript for publication, providing that you modify the manuscript according to the review recommendations. 

Due to the previous editor being currently unavailable, I have been assigned as the editor for the revised version of your manuscript.

While reviewing the revised manuscript, I have concluded that all comments and criticisms presented by the reviewers have been correctly addressed. 

However, I have also noticed that in Fig. 5A, the graph shows that the miRNA mimic increased luciferase activity, while the text indicates that luciferase activity was decreased by the RNA mimic (as would be expected, given the rest of the results shown in Figs. 5 and 6). Please check if there was an error during the preparation of Fig. 5A, or revise the text and conclusions if necessary.

Additionally, in Fig. 5A the inset reads "Mimcs" instead of "mimic", and the NC mimic is not described in the text or figure legend.

Finally, please check l. 258, "hepatocytes (Alb-positive cells) were significantly decreased (Fig 2B and S2 Fig).", as the marker for hepatocytes used in Fig. S2 was cytokeratin 18, and not albumin.

Sincerely,

Uriel Koziol

Section Editor

Eva Clark

Section Editor

Due to the previous editor being currently unavailable, I have been assigned as the editor for the revised version of your manuscript.

While reviewing the revised manuscript, I have concluded that all comments and criticisms presented by the reviewers have been correctly addressed. 

However, I have also noticed that in Fig. 5A, the graph shows that the miRNA mimic increased luciferase activity, while the text indicates that luciferase activity was decreased by the RNA mimic (as would be expected, given the rest of the results shown in Figs. 5 and 6). Please check if there was an error during the preparation of Fig. 5A, or revise the text and conclusions if necessary.

Additionally, in Fig. 5A the inset reads "Mimcs" instead of "mimic", and the NC mimic is not described in the text or figure legend.

Finally, please check l. 258, "hepatocytes (Alb-positive cells) were significantly decreased (Fig 2B and S2 Fig).", as the marker for hepatocytes used in Fig. S2 was cytokeratin 18, and not albumin.

Figure Files:

Data Requirements:

Reproducibility:

References

---

## [Editor Report · Decision Letter 4]

8 Jul 2023

Dear Dr Guo,

We are pleased to inform you that your manuscript 'mmu-miRNA-342-3p promotes hepatic stellate cell activation and hepatic fibrosis induced by Echinococcus multilocularis infection via targeting Zbtb7a' has been provisionally accepted for publication in PLOS Neglected Tropical Diseases.

Best regards,

Uriel Koziol

Section Editor

Eva Clark

Section Editor

---

## [Editor Report · Acceptance letter]

20 Jul 2023

Dear Dr Guo,

We are delighted to inform you that your manuscript, "mmu-miRNA-342-3p promotes hepatic stellate cell activation and hepatic fibrosis induced by *Echinococcus multilocularis* infection via targeting Zbtb7a," has been formally accepted for publication in PLOS Neglected Tropical Diseases.

Best regards,

Shaden Kamhawi

co-Editor-in-Chief

Paul Brindley

co-Editor-in-Chief
